# The Consumption of Insects in Switzerland: University-Based Perspectives of Entomophagy

**DOI:** 10.3390/foods11182771

**Published:** 2022-09-08

**Authors:** Aline Oliveira Penedo, Sophie Bucher Della Torre, Franziska Götze, Thomas A. Brunner, Wolfram Manuel Brück

**Affiliations:** 1Geneva School of Health Sciences, HES-SO University of Applied Sciences and Arts Western Switzerland, Rue des Caroubiers 25, 1227 Carouge, Switzerland; 2Food Science & Management, School of Agricultural, Forest and Food Sciences (HAFL), Bern University of Applied Sciences, Länggasse 85, 3052 Zollikofen, Switzerland; 3Institute of Life Technologies, School of Engineering, HES-SO University of Applied Sciences and Arts Western Switzerland Valais-Wallis, 1950 Sion, Switzerland

**Keywords:** insects, entomophagy, consumer acceptance, sustainable foods

## Abstract

Although insects have long been part of the human diet in many countries, they are poorly received and accepted in European and North American countries. Therefore, this cross-sectional observational study, based on a structured questionnaire, aimed to evaluate the level of acceptability of entomophagy among young adults in a Swiss university context. The variable “acceptability of consuming insects” (ACI) was calculated according to the perception of entomophagy of each participant. The ACI was related to various socio-demographic and behavioral aspects. A total of 290 responses were validated and analyzed. The mean ACI score was 3.7 out of 6.0 (SD 1.1). Most participants responded that the most likely reason for eating insect foods was curiosity. The most common reason for not eating such foods was disgust. None of the socio-demographic variables showed a significant association with ACI. Generally, participants in this study showed a potential interest in entomophagy—on a theoretical level, as measured here by the ACI. In practice, however, there are still barriers, including disgust, which contribute to the low consumption of these foods, at least in Switzerland.

## 1. Introduction

Entomophagy is defined as the practice of eating insects [1]. In many cultures, particularly in the sub-Saharan Africa, south and southeast Asia, and Latin America, insects have been consumed for centuries [2,3]. As such, more than 1900 species of insects have already been used as food with insects being part of the daily diet of at least two billion people [1,4,5,6]. Among the vast number of edible insect species are beetles, caterpillars, bees, wasps, ants, grasshoppers, locusts, and crickets [5].

Edible insects are generally valued for their high energy content and the quality of their protein, thanks to an amino acid profile that can match human nutritional requirements. For some species, the proportion of protein can exceed 20 g per 100 g portion, which is equivalent or even higher than chicken breast. Despite the long history and nutritional benefits of entomophagy, it was only in 2013, following a major publication by the Food and Agriculture Organization of the United Nations (FAO), that discussions about this practice and its possible role in global food security expanded, especially in Europe and North America [1,7]. As a result, insects are now seen as one of the future pillars of a sustainable human diet.

Compared to other alternative protein sources, the advantage of insects is their low soil requirement for rearing, which allows for an intensive production in small areas [8]. Indeed, vertical farming of insects may greatly optimize the space used for their rearing [9]. Insect farming also requires much less water, compared to the rearing of other animals for food [2]. Therefore, the production of insects as foods has a better feed conversion ratio and generates less greenhouse gases [10]. For example, to produce 1 kg of protein, mealworms need only 10% of the land required for beef production [9]. Crickets require only 1.7 kg of food for every 1 kg of their body weight to grow [11]. Moreover, insect food production carries a low risk of zoonotic disease transmission—a strong argument in the wake of the COVID-19 pandemic [4,11,12,13].

However, the current global scale of insect production is insufficient to be considered a sustainable source of nutrients. Thus, to establish entomophagy and its potential role in reducing the environmental impact of the food industry, the European Union has adopted the Novel Foods Regulation in 2015 (Regulation (EC) No 2015/2283), allowing companies to market innovative foods and ensuring that these same companies meet advanced food safety and quality standards for all European consumers [14]. Similarly, in 2017, the Swiss Federal Office for Food Safety and Veterinary Affairs (FSVO) published technical guidelines on official controls in primary insect production, as well as an information sheet containing a description of insects authorized as foodstuffs, the conditions for their production and release [15]. Three species of insects have been approved as foodstuffs in Switzerland, namely: (i) *Tenebrio molitor* in the larval stage (yellow mealworm); (ii) *Acheta domesticus*, adult form (house cricket); and (iii) *Locusta migratoria*, adult form (migratory locust).

Although legislation is being passed, consumer acceptance of insect-based foods is being investigated internationally. Studies generally show that most Western consumers are not yet ready to consume insects [16]. Indeed, a recent study found that there is the need for caution regarding the prospects of edible insects as the “food of the future” [17]. The consumers surveyed responded with a natural aversion to insects as foods [10,18,19]. This food neophobia is, therefore, an important factor to consider in the creation of new, insect-based foods [16]. Consumer food neophobia, together with perception of disgust, has been shown to significantly influence the individual decision to try insect-based foods. Thus, these are important predictors that contribute to the acceptability of new food products [4,20,21].

From a socio-cultural point of view, one of the beliefs associated with insect consumption is that only economically disadvantaged groups adhere to it, especially in conditions of food scarcity [22]. Indeed, preconceived notions about insect foods, as well as lack of knowledge about their benefits, are factors associated with low consumer acceptability of eating insect foods [2]. In a survey with 542 participants, Schlup and Brunner observed that only a small segment of the Swiss population (9.3%) was in favor of eating insects [23]. Swiss consumers who were open to entomophagy had a high level of education and scored low on the food neophobia scale. However, only one in three people reported having ever eaten insects. These people showed high nutritional knowledge and had read about entomophagy in the media. These results show that in Switzerland this type of food was still considered a modern and innovative product in 2015. Another study found that, in Switzerland, people who consumed insect products (e.g., insect burgers) were perceived as more health-conscious and environmentally friendly, amongst other qualities, than meat consumers [24]. In view of this positive image, it is possible that group dynamics influence the eating behavior of the Swiss population.

In Switzerland, there is a need to better understand the aspects that determine the willingness to consume insect-based foods so that they can, in fact, be integrated into an overall sustainable food strategy. There is a lack of studies on the acceptability of entomophagy following recent changes in the regulatory framework, which have potentially impacted the supply of products [15]. On this basis, the aim of this study was to better understand the viability of insect-based food products in Switzerland from the perspectives of the younger generation of consumers. In this sense, the study proposed to evaluate the level of acceptability of entomophagy among young adults in a Swiss university context, considering different socio-demographic and behavioral aspects. This study aimed to answer the following research questions:Question 1: What proportion of the target population is potentially interested in eating insect-based foods, based on environmental, nutritional, economic, gastronomic, and cultural arguments?Question 2: What are the facilitating and inhibiting factors for the consumption of insect-based foods in the studied population?Question 3: What socio-demographic and behavioral aspects are associated with the acceptability of consuming insect-based foods?

## 2. Materials and Methods

The present work is a cross-sectional observational study aimed at assessing the acceptability of eating insect-based foods among young adults enrolled at a university or college in Switzerland, using a structured questionnaire. In the study, “acceptability of consuming insects” was used as a dependent variable, which was then related to socio-demographic and behavioral aspects (independent variables).

### 2.1. Definition of Acceptability (Dependent Variable)

For the definition of the dependent variable, this work was based on a similar study conducted in Switzerland by Schlup and Brunner [23]. In their study, the authors used the term willingness to consume to refer to the acceptability of participants to eat insects. Here, we will use the term ‘acceptability of consuming insects’, henceforth referred to as ‘ACI’. Like Schlup and Brunner, the ACI was measured based on 5 questions (see questions Q1 to Q5 in the questionnaire, Appendix A). These questions assessed the level of agreement or disagreement of the participants with arguments in favor of entomophagy, covering 5 themes (sustainability, health, price, taste, and trend), according to a Likert scale ranging from 1 to 6 (1 = strongly disagree; 6 = strongly agree). This six-point scale is identical to that of Schlup and Brunner and was chosen because it ensures that an opinion is obtained on each question, avoiding “neutral” responses [23]. Using the 5 questions mentioned above, it was possible to assign each participant a score in the continuous range of 1.0 to 6.0 corresponding to their ACI. An ACI score of 3.5 (mean value of range between 1 and 6) or higher (an individual score of either 4, 5, or 6) was interpreted as potentially indicating an interest in eating insect-based foods (and vice versa for values lower than 3.5). The calculation of the ACI score is explained in the statistical analysis section.

### 2.2. Target Population and Inclusion Criteria

The study focused on the population of students enrolled at Swiss-based universities and universities of applied sciences. This choice of target audience was made based on the study’s access to the participating population. This was facilitated by the availability of emails from people studying and working at the universities, as recruitment for participation in the questionnaire was done by sending emails and invitations to participate via the universities’ social media.

Younger adults tend to be more sensitive to health issues and more sustainable in their food choices than older people, who are more resistant to changes such as accepting new foods [24]. Furthermore, compared to adolescents, young adults are at a point in their lives where they are beginning to have the opportunity to make more conscious and independent food choices, with more opportunities to explore new foods [16]. Most importantly, today’s youth are tomorrow’s consumers, which makes understanding their food preferences particularly interesting for assessing the viability of insect-based foods.

### 2.3. Recruitment and Exclusion Criteria

Participants were recruited by an e-mail invitation sent to the French-, German-, and Italian-speaking regions in Switzerland. Different databases containing the e-mail addresses of students enrolled at a university or university of applied sciences in Switzerland were used for recruitment. The email contained a direct link to the survey, which was developed on the Google Forms platform. An e-mail address was provided in case participants wished to send a comment or make a remark about the study.

Data collection took place over three months, from July to September 2021. The questionnaire was first sent out in July 2021, and then a reminder was sent out at the start of the academic year in September 2021—thus reaching the minimum number of participants for the study. In total, the recruitment email was sent to over 1500 people.

It is possible that the email was informally forwarded by the recipients themselves, and that more people received the invitation without necessarily being part of the target audience. In addition, the databases used did not exclusively contain students, but anyone affiliated with a university or college in Switzerland. In view of these two aspects, it was necessary to sort the participants to exclude those who were not part of the target audience, i.e., those not being students in Switzerland.

An exclusion criterion was therefore set up, based on age: all participants older than 36 years were excluded. This threshold was determined by considering the age of a person at the end of their PhD. In addition to the age, location (Switzerland) and presence at a university of university of applied sciences, no other exclusion criteria such as health status or diet were established.

### 2.4. Questionnaire

The questionnaire used in this study is an adaptation of the one developed and validated by Schlup and Brunner [23]. A digital version of the questionnaire was constructed using the Google Forms platform. To avoid incomplete questionnaires being returned, all questions were mandatory so that each participant was required to answer all questions to submit their questionnaire. For the sake of inclusiveness, the questionnaire was distributed in French, German, and Italian (three of the Swiss national languages). Before being distributed, the adapted questionnaire was tested with a convenience sample. The English version of the questionnaire is provided in Appendix A.

The questionnaire was structured as follows:Section 1: arguments in favor of insects (5 questions, Q1 to Q5 of the questionnaire);Section 2: eating behavior (4 questions, Q6 to Q9 of the questionnaire);Section 3: socio-demographic data (1 question, Q10 of the questionnaire, broken down into 6 points).

As mentioned above, the first section addressed the arguments in favor of consuming insect-based foods, through five themes: sustainability, health, price, taste, and trend. The answers to these five initial questions were used to calculate each participant’s ACI score. The second section looked at cultural and behavioral issues, such as participants’ values and beliefs around entomophagy, meat consumption, and reaction to new types of food. This section also looked at the presentation of insect-based foods to understand how this may impact their acceptability to participants. This considers recent studies that show a correlation between the acceptability of consuming insect-based foods if they are visible as a whole or non-visible (processed) on the plate [23]. The third section dealt with socio-demographic aspects considered relevant in the literature, such as gender, age, country of origin, or linguistic region [25,26,27].

Except for Section 3 (socio-demographic data), for all questions participants were asked to respond on a Likert scale. The scale ranged from 1 (“strongly disagree” or “very unlikely”) to 6 (“strongly agree” or “very likely”). The intermediate values were anchored in numerical form only [4,20,28].

### 2.5. Statistical Analysis

Statistical analyses were carried out using STATA (version 16.1) and RStudio (version 2021.09.0) software. First, a purely descriptive analysis was carried out to obtain the profile of the participants, based on the following variables: gender, age category, linguistic region, current education level, main field of study, and nationality. This allowed for the identification and exclusion of all participants over the age of 36.

Then, to answer research question 1, each participant’s ACI score was calculated by averaging the answers to the first 5 questions (Q1 to Q5 of the questionnaire, see Appendix A), resulting in a continuous numerical value between 1.0 and 6.0. As mentioned above, participants with an ACI score of 3.5 or higher were considered potentially interested in eating insect-based foods. This threshold (3.5) corresponds to the middle value of the interval given by the Likert scale. It should be noted that, as in the study by Schlup and Brunner, the ACI is in fact a theoretical acceptability, defined according to the affinity of the participants with the arguments presented in favor of entomophagy—thus being measured independently of whether the participants had previously consumed insect-based foods [23]. To check the relevance of the ACI score in our questionnaire, it was compared with the participants’ previous experiences, who were asked to specify whether they had ever eaten such foods.

To answer research question 2, the analysis focused on the answers obtained in the second section of the questionnaire (eating behavior). It was the answers to question 7 (broken down into 3 points, i.e., Q7.1, Q7.2, and Q7.3) that were relevant at this stage of the analysis. Thus, to highlight the facilitating or inhibiting factors to the consumption of insect-based foods among the participants, an analysis of the frequency of responses to questions 7.2 and 7.3 was carried out. This made it possible to identify: (i) what was the most frequent reason for not eating insect-based foods among those who have never eaten them, and (ii) what was the most frequent reason for eating insect-based foods among those who have eaten them.

To answer research question 3, the ACI score was related to the behavioral and socio-demographic variables identified in Sections 2 and 3 of the questionnaire. This was done through Chi-square and Fischer tests, where the ACI score was used as the dependent variable, and the behavioral and socio-demographic variables as independent variables. This time, the ACI was treated as a discrete variable: for each participant, its value was rounded to an integer between 1 and 6 (thus generating 6 categories). This made it possible to compare the ACI with the various socio-demographic categories. Here, it should be noted that, given the low response from the Swiss Italian population, it was decided to group Italian and French speakers into a single group (Latin Swiss). The behavioral variables, on the other hand, required a similar pre-processing to the ACI. First, for each participant, the averages of the responses to different groups of questions (each group being linked to a specific behavioral aspect) were calculated, giving continuous numerical values between 1.0 and 6.0. Like the ACI, these values were rounded to discrete variables (integer 1–6). In this way, the behavioral aspects could be inserted as categorical variables in the Chi-square and Fischer models to test their association with the ACI. The accepted level of significance for testing the null hypothesis was set at *p* < 0.05.

### 2.6. Internal Consistency and Cronbach’s Alpha

As mentioned above in Section 2 (questions about eating behavior in general), different responses were aggregated to form behavioral variables. All these responses followed a scale from 1 to 6; however, the direction of the scale was not constant, being sometimes reversed: for example, in one question the value 6 corresponded to “I constantly taste different or new foods”; while, in another question, 6 corresponds to “I do not trust new foods”. As a result, some questions had to be reverse coded before being translated into a discrete variable (as described above in the statistical analysis). The use of reversed statements allows for testing the overall consistency of each participant’s responses.

To assess the internal consistency of the questionnaire, Cronbach’s alpha (α) was calculated for each variable derived from the aggregation of responses to 2 or more questions—i.e., the ACI score and the behavioral variables. This allowed for a check of the overall consistency of the responses composing each variable according to the Likert scale. Based on other studies, the threshold for Cronbach’s alpha consistency was set at α ≥ 0.70 [29].

### 2.7. Ethical Considerations

In this study, no data on the health of the participants were collected. Therefore, it does not fall within the scope of the Federal Act on Research Involving Human Beings and was, therefore, not submitted to an ethics committee for approval.

At the beginning of the questionnaire, an introductory section informed the participants about their contribution to this research, as well as about the confidentiality of their answers. Participation in the study was voluntary and responses were treated as confidential. All data were anonymized during processing (through aggregation processes), so that no answers could be linked to a specific person participating in the study. Finally, participants did not receive any compensation.

## 3. Results

### 3.1. General Information about the Participants

A total of 405 people responded to the questionnaire. Of these participants, 115 did not meet the inclusion criteria (people over 36 years old). This resulted in a total of 290 valid responses which met the quantitative requirements of the study.

### 3.2. Characteristics of the Participants

A summary of the respondents’ gender, age group, linguistic region, level of education, main field of study, and nationality is given in Table 1. Overall, 57% of participants were female, with the largest proportion of participants being in the field of Health Sciences (32%); followed by Natural, Computer and Life Sciences (24%); and Engineering and Architecture (14%). Respondents aged between 18 and 24 were the most numerous, representing 49% of the sample. Most of the study population were currently in Masters’ degree programs (37%), followed by Bachelor degree students (33%). In terms of language region, most participants (73%) reported living in Latin Switzerland (Suisse Romande or Ticino). Overall, 80% of the participants were of Swiss nationality.

### 3.3. Acceptability of Eating Insect-Based Foods

To meet the first objective (determine the proportion of the study population that would be willing to eat insect-based foods), we calculated the ACI score. Figure 1 and Table 2 show that the mean ACI score (represented by the red line) was 3.7 (with a standard deviation of 1.1) and the median was 3.8 (orange line). The ACI appears to follow a normal distribution with the number of responses decreasing as the score approaches the extremes (1 and 6). The largest proportion of responses is concentrated in the range between 3.5 and 3.9, with 22.1% of participants having an ACI score in this range.

In total, 62.5% of the participants had an ACI score of 3.5 or higher. Most participants were, therefore, potentially interested in entomophagy. As mentioned above, the ACI was reported in a theoretical dimension, namely the participants’ level of agreement with different arguments in favor of eating insect-based foods. To check the relevance of the ACI score, it was compared with the participants’ previous experiences with eating this type of food (Table 3).

More than half of the respondents had already consumed insects (55%). However, of these, most had done so only once (87 out of 160). It is noted that the average ACI scores increase with the reported frequency of insect food consumption (average ACI of 3.4 among those who have never eaten insects compared to an average ACI score of 4.4 among those who have eaten them several times).

Table 4 shows the results in more detail and highlights the differences in responses for men and women. Of those who had never eaten insects, women made up the vast majority (*n* = 90). A total of 50 women reported having eaten insect food once, compared to 37 men. For the most frequent consumption, only 25 women responded that they had eaten insect food several times, compared to 46 men.

### 3.4. Facilitating and Inhibiting Factors for the Consumption of Insect-Based Foods

To meet the second objective of the study (identify the facilitating and inhibiting factors for the consumption of insect-based foods among the target population) the analysis focused on the participants’ previous consumption. These analyses were performed according to two subgroups: (1) people who had never eaten insects and (2) people who had already eaten insects (once or several times).

First, to better understand what reasons would encourage people who have never eaten insect food to do so, response options were given on different aspects that could be considered (Figure 2). Of the 128 respondents who had never eaten insect food, half (*n* = 64) stated that the most likely reason for them to eat such food would be curiosity. Of these, 45 (70%) were women. The category ‘other reasons’ ranked second (*n* = 35), followed by ‘because eating insects is sustainable’ (*n* = 9).

The same question was analyzed among the second group, i.e., those 158 people who had already eaten insect-based food once or several times (Figure 3). This made it possible to identify the most common reason for eating insect food. Here, as well, curiosity was the most frequently mentioned reason, being present in most responses (*n* = 135, or 85% of the total).

Answers to the question: “Why have you never eaten insects before, or why don’t you eat them more often?” are shown in Figure 4. To this question, almost 40% of the participants answered that eating insects is disgusting. To this question, most responses were given by female participants (73%). In addition, 27% said they had not had the opportunity to eat or had not found these foods, while 8% said they preferred “real” meat.

### 3.5. Acceptability by Socio-Demographic and Behavioral Variables

As shown in Table 5, none of the socio-demographic variables were significantly associated with the ACI. In Table 6, the following variables concerning eating habits were significantly (positively or negatively) associated with the ACI: “Foods with recognizable insects”, “Foods with non-recognizable insects”, “Eating meat is synonymous with health”, “Food neophobia”, and “Neophobia to new food technologies”.

## 4. Discussion

### 4.1. Acceptability of Eating Insect-Based Foods

To find out what the proportion of the study population that was interested in ACI was, the threshold of acceptability proposed for the analysis was the central value of the interval corresponding to the Likert scale (1–6), i.e., 3.5. This means that 62.5% of the participants would be potentially interested in entomophagy. A more restrictive threshold, where it would be even more certain that participants would accept entomophagy, could also have been used; for example, a threshold of 4.0 (40.4%) or even 5.0 (9.7%). Indeed, it is difficult to define a threshold in the spectrum of acceptability, based on arguments and not on practice (because of the intention-behavior gap). In order to contextualize these results, the scores of the variables composing the ACI were compared with those obtained in the previous study of Schlup and Brunner, who used exactly the same five arguments to calculate a score comparable to the ACI (also ranging from 1 to 6), which they named “Willingness To Consume” (WTC) [23]. This comparison is shown in Table 7.

Comparing the results obtained in this study with those of Schlup and Brunner, the difference between the values of the means relating to the first four arguments (sustainability, health, price, and taste) were highly significant (*p* < 0.001) [23]. These same arguments scored higher in the present study than in that of Schlup and Brunner, suggesting a possible evolution of the customs of recent years regarding the acceptability of eating insect-based foods [23]. More specifically, the “level of agreement” for the first four arguments had means above 4 for this study, being systematically higher than in the reference study. In terms of trend, however, no significant differences were found in the responses.

Considering that Schlup and Brunner’s study was conducted in 2015, just before the changes in the regulatory framework in Switzerland on the sale and marketing of insect products, people’s acceptability of entomophagy has evidently changed over the last 6 years. As mentioned above, these legislative changes may have had a positive impact on the supply and demand for this type of product. It probably encouraged consumers to try this new food that was appearing on the shelves of mainstream supermarkets. However, there is also a possible population bias, as in this study we focused on the young adult population who are probably more open-minded towards novel foods such as insects than the rest of the Swiss population. This contrasts with the study by Schlup and Brunner where age had a marginally negative effect on WTC in the general Swiss population [23].

Despite the differences between the means, the relative importance of the arguments did not change between the two studies. Sustainability remains the most accepted argument, while the questions related to taste and tendency to consume insects showed the most disagreement. In both studies, taste received a score close to the central value of the scale (of 3.5), which means that these arguments for consuming insects are still relatively weak or that respondents had not formed a strong opinion on these issues.

As presented in the article by Schlup and Brunner, there is an apparent disconnect between the perceived relevance and the actual influence of the argument on the participants’ decision to consume insects [23]. It is perceived that positive arguments allow participants to become aware of the benefits of insect consumption. However, certain obstacles to the effective consumption of this type of food remain. This is illustrated by the proportion of participants with a potential interest in entomophagy (ACI ≥ 3.5, i.e., 62.5%), and the proportion of participants who have consumed insect-based foods more than once (25.2%).

### 4.2. Factors Motivating or Inhibiting Entomophagy

In this study, curiosity was the main reason why people might consume insect-based foods. Curiosity is indeed a factor that motivates people to try new types of food [2]. Curiosity about new foods is often aroused in a group dynamic. Studies have shown that people are more willing to try new foods such as insect-based foods in a relaxed setting with friends, for example, as an aperitif, given the variety of insect-based chips and burgers [20]. One can also consider “peer pressure” as a factor among people to at least accept to taste this type of food [30,31]. However, peer pressure can also have the opposite effect, triggering reactions ranging from disgust to food neophobia—depending on the group dynamic [32]. Sustainability was of rather marginal importance as a reason for participants to eat insect-based foods. Indeed, while participants showed a high level of agreement with the environmental arguments presented at the beginning of the questionnaire, the same participants only rarely mentioned sustainability as the main reason for eating insects. Interestingly, sustainability was more frequently cited as a reason among those who had never eaten insects than among those who had. This suggests that the awareness for climate change may be changing eating habits.

### 4.3. Association between ACI and Socio-Demographic Variables

To investigate whether acceptability was associated with socio-demographic variables, this study conducted Chi-square and Fisher tests. Notably, none of the selected socio-demographic variables (gender, age category, language region, current level of education, main field of study, and nationality) were significantly associated with ACI. These results contrast with previous studies where it was shown that gender shows a significant association with acceptability (females/males are more/less likely to …), while other studies show that age and education level have significant associations (explanation …) [20,27,33].

According to recent studies, gender seems to impact the acceptability of alternative proteins. In the case of insects, they are generally better accepted by men [16,34,35,36,37]. Looking at the distribution of the ACI score, this association was not found in the present study. However, this result can be qualified by the data concerning previous insect consumption (which was not considered in the ACI). Of the 25% of participants who had consumed insects more than once, men were in the majority (46 out of 71, or 65%), whereas women were in the majority among all participants (57.7% of the total). It is true that women are more attentive to healthy eating and are, therefore, more open to new foods with a positive impact on health [38]. However, studies suggest that this same audience has a strong sense of disgust and high levels of food neophobia towards insect-based foods [32]. This was also observed in the present study, as disgust was one of the main reasons that inhibits insect consumption (Figure 4). At the same time, men were less reluctant and more open to eating foods with a perceived “disgusting” appearance. Indeed, studies show that their threshold for food neophobia is generally higher than that of women [39].

Another result that contrasts with the existing literature is the lack of a significant association between age and ACI. This may be explained by the small differences in the age ranges of the participants, who were all relatively young. In a study in Germany, young people were open to entomophagy than older individuals [35]. Another study in Belgium found that a 10-year age difference in age was associated with a 27% decrease in the likelihood of participants being willing to adopt insects as food [39].

For the other variables, limitations related to the representativeness of the sample certainly biased the results. The level of education lacked contrast, being high for all participants (university or university of applied sciences). The nationalities of the participants were varied, but the non-Swiss remained few. Finally, the linguistic regions were not well represented, considering the prevalence of French speakers and the low number of Italian speakers.

### 4.4. Association between ACI and Behavioral Variables

In the present study, the majority (5 out of 9) of the behavioral variables, i.e., (i) food with visible insects, (ii) foods with non-visible insects, (iii) meat is synonymous with health, (iv) food neophobia, and (v) neophobia to new food technologies, were significantly (*p* < 0.05) associated with ACI. These results are consistent with the literature.

Regarding the first two variables (insect visibility), recent studies have shown increased acceptability when the insect is processed, e.g., as flour, and an ingredient of, for example, baked goods, pasta, brownies, crisps, or protein bars [40]. A study conducted in Switzerland in 2015 aimed at understanding the emotional experiences of insect-based snacks confirmed the impact of processing on consumer perception, showing that the more processed the insect-based food is, the more positive the emotional evaluation of the product is [41]. Furthermore, the authors considered that repeated exposure is necessary to accustom consumers to these new foods. They thus reveal the role of marketing and the supply of these foods in the market: with the introduction of new products, closer to traditional products, emotional barriers can be overcome and the willingness to consume insects increases. This impact of the product offer on the perception of the increase in the acceptance of insect-based foods can be seen by comparing the results of the present study with those of Schlup and Brunner: the number of people who reported having eaten insects (at least once) increased from 18.9% in 2015 to 55% in 2021 (present study) [23]. The new Swiss regulatory framework (established in 2017) and the evolution of the product offer taking place during these six years have likely contributed to increase the acceptability of these products among the Swiss population.

Regarding the variables linked to meat consumption, according to a qualitative study carried out in Switzerland, the increase in the supply of various sources of protein leads to an equally varied consumption of food products [42]. However, meat consumption seems to be very present among the participants of the present study: 15% of the participants declared that they did not consume meat, 7% consumed it less than once a month and a little more than half consumed it 2 to 4 times a week. It is likely that these people are aware of the need to reduce meat consumption and take advantage of new food technologies to discover meat substitutes, such as insects [43].

It should be noted that the variable “meat lovers” was not significantly positively or negatively associated with ACI, whereas the variable related to neophobia of food technologies (techneo) was. This result shows that liking meat is not associated with greater acceptability (or rejection) of insect food consumption. Alexander et al. (2017) argue that gradual (non-disruptive) changes in consumer eating habits should be induced, facilitating a transition to more sustainable diets [44]. Indeed, meat and insect consumption are not mutually exclusive. Insects can gradually be inserted (in different forms) onto our plates as food technologies advance.

Although meat consumption was not clearly associated with ACI, another diet-related factor was relevant to understanding the refusal of entomophagy. Some people who received the recruitment email refused to participate, stating that they were vegans or vegetarians and did not agree to participate in this type of research.

### 4.5. Limitations of the Study

This study has limitations related to the convenience sampling method, which is not random and may have selection biases. For example, the percentage of women (57.7%) and men (42.3%) differed slightly from the proportions observed in Switzerland in 2020, which were 50.4% and 49.6%, respectively [45]. This discrepancy can certainly be explained by the age range and field of study of the target population. For example, according to the Swiss Federal Statistical Office, in 2018, women were in the majority among young doctors, accounting for 62% of doctors aged 35–39 years [46]. This may explain the high prevalence of women among the participants, as the questionnaire reached more people whose main field of study is health sciences.

Moreover, in studies with voluntary sampling (such as this one), it is possible to produce a bias that could overestimate the real consumption because the participants who agree to answer the questionnaire are more concerned or who identify with the topic being studied.

In addition, considering that the authors had greater access so the universities in the French-speaking part of Switzerland, there was an increased probability that an above-average proportion of the participants would come from this linguistic region. For the same practical reasons, most of the participants were Swiss. This makes it difficult to extrapolate the results of this study beyond Western Switzerland. In a future study, the questionnaire could be offered in more languages, for example. In addition, the recruitment process could be expanded, for example, by recruiting participants from the various local universities on site.

## 5. Conclusions

In a context of climate emergency and increasing limitations in terms of arable land availability, it is essential to reassess the nutritional potential and possible role of insects in global food security. The first step is to assess the acceptability of insect-based foods to consumers, particularly in the West where there is no tradition and habit toward them yet.

This study investigated the perception of entomophagy among young adults in a Swiss university context. Overall, the theoretical arguments in favor of entomophagy were well accepted. In practice, however, the study participants were less inclined to consume insect-based foods. Nevertheless, the overall acceptability of these foods, as well as the number of people who have eaten insects at least once, appears to have increased between 2015 and 2021. This has probably been influenced by the regulatory framework put in place in Switzerland since 2017, which has helped to expose consumers to this new product and, thus, to attract their attention.

Significantly, among the participants in this study, curiosity was by far the most mentioned motivating factor for eating insect-based foods. Whereas no socio-demographic variables were found to be significant in predicting theoretical acceptability (ACI), five behavioral variables were significantly related to ACI: (i) food with visible insects, (ii) food with non-visible insects, (iii) meat is synonymous with health, (iv) food neophobia, and (v) neophobia to new food technologies. These results suggest that there is still resistance to these products due to factors, such as presentation (level of processing) and food neophobia.

Given the trends discussed in this study, it is reasonable to assume that in the coming years entomophagy will expand in Switzerland. However, entomophagy still needs to be better considered by different sectors: research, health, food industry, associations, and governments (through the regulatory framework of these foods). There is also a need for further research into the benefits of insects as a food and as a potential substitute for meat, so that they can be used as arguments in marketing to consumers.

## Figures and Tables

**Figure 1 foods-11-02771-f001:**
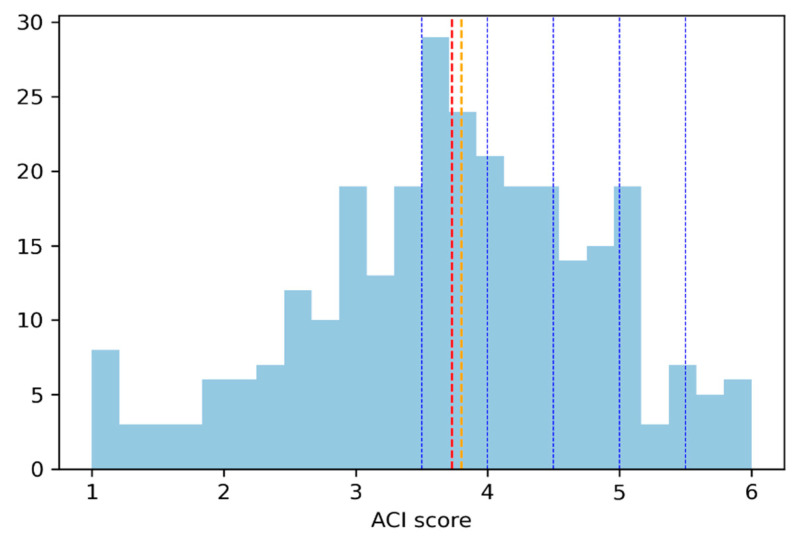
Distribution of ACI scores, by intervals. Red line: the mean of ACI score was 3.7; Orange line: the median was 3.8; Blue lines mark the ACI score intervals are shown in Table 2.

**Figure 2 foods-11-02771-f002:**
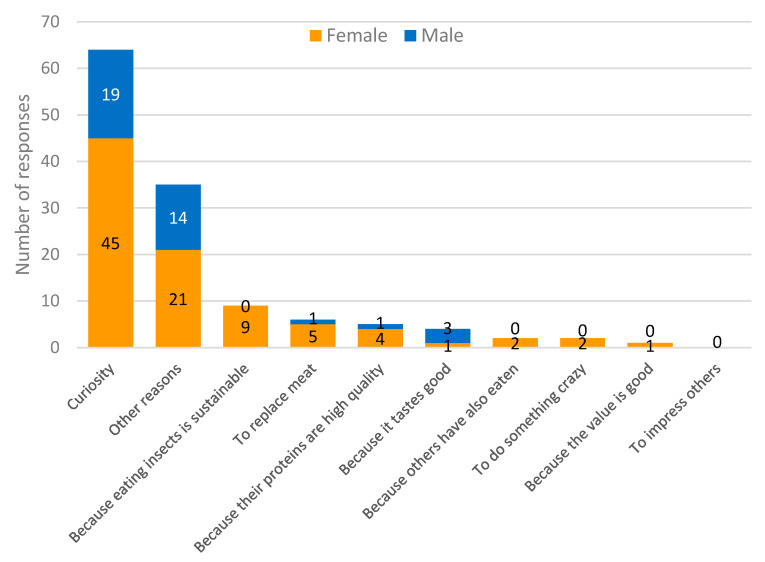
Most likely reason for participants who have never eaten insect-based foods to try such foods.

**Figure 3 foods-11-02771-f003:**
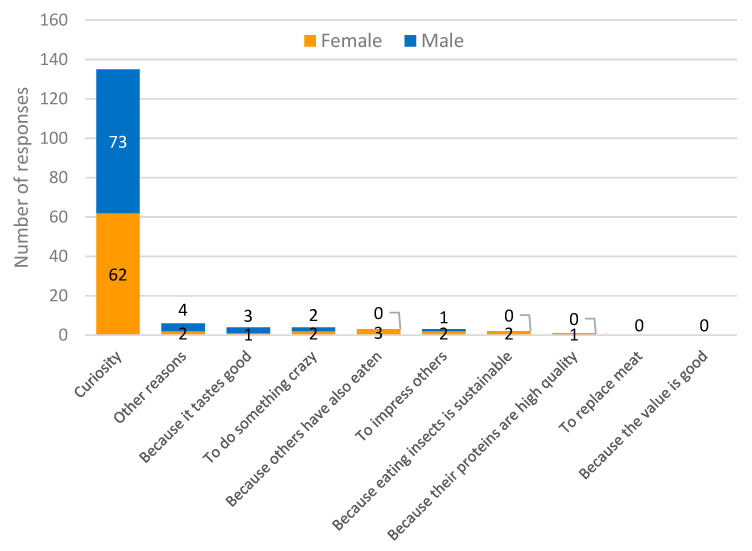
Most likely reason for eating insect food among those who have ever eaten it.

**Figure 4 foods-11-02771-f004:**
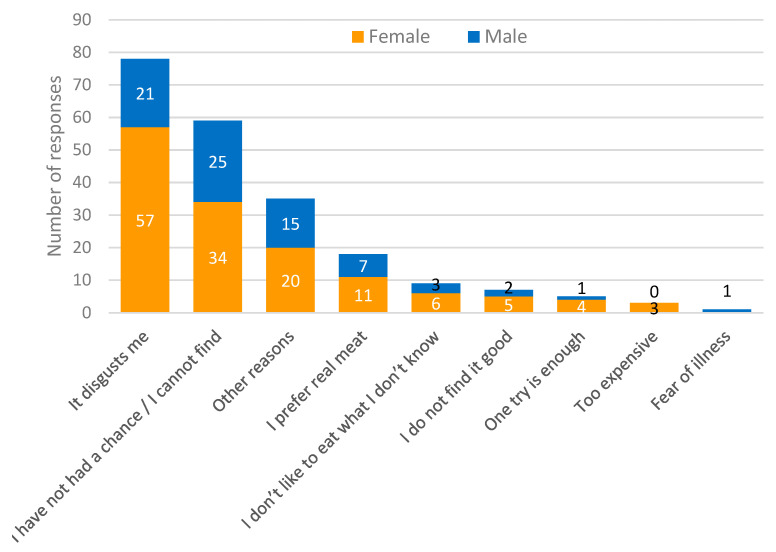
Reasons for never having eaten insect food or having eaten only once.

**Table 1 foods-11-02771-t001:** Socio-demographic characteristics of the sample.

Features	*N* = 290
** *Type* **	
Female	165 (57%)
Male	121 (42%)
I prefer not to answer	4 (1.4%)
** *Age Category* **	
18–24	142 (49%)
25–30	106 (37%)
31–36	42 (14%)
** *Language Region* **	
Latin Switzerland	211 (73%)
German-speaking Switzerland	79 (27%)
** *Current Level of Education* **	
High School Diploma	49 (17%)
Bachelor	97 (33%)
Master	108 (37%)
PhD	11 (3.8%)
Other	25 (8.6%)
** *Main Field of Study* **	
Health Sciences	94 (32%)
Natural sciences, computer sciences, and life sciences	71 (24%)
Engineering and architecture	40 (14%)
Economics, trade, management, and services	26 (9.0%)
Other	25 (8.6%)
Humanities and social sciences	24 (8.3%)
Pedagogy and educational sciences	6 (2.1%)
Law	4 (1.4%)
** *Nationality* **	
Switzerland	231 (80%)
France	23 (7.9%)
Brazil	8 (2.8%)
Portugal	7 (2.4%)
Germany	5 (1.7%)
Spain	5 (1.7%)
Other	11 (3.5%)

**Table 2 foods-11-02771-t002:** Distribution of ACI scores, by intervals.

ACI Score	Proportion (%) of Participants with ACI Score within Given Interval
3.5–4.0	22.1
4.1–4.5	16.6
4.6–5.0	14.1
5.1–5.5	5.9
5.6–6.0	3.8

**Table 3 foods-11-02771-t003:** Previous experiences with insect food consumption and respective ACI scores.

Have You Eaten Insect-Based Foods?	*N*	%	Average ACI
No, never	130	44.8%	3.4 (SD = 1.1)
Yes, once	87	30.0%	3.8 (SD = 1.0)
Yes, several times	73	25.2%	4.4 (SD = 0.9)
TOTAL	290	100.0%	3.7 (SD = 1.1)

**Table 4 foods-11-02771-t004:** Previous experiences with insect food consumption, by gender.

Have You Eaten Insect Food (M/F)	Type	*N*	%
No, never	Female	90	31%
	Male	38	13%
Yes, once	Female	50	17%
	Male	37	13%
Yes, several times	Female	25	9%
	Male	46	16%
Total, excluding gender not declared (*n* = 4)		286	100%

**Table 5 foods-11-02771-t005:** Acceptability of consuming insect-based foods by socio-demographic variables.

		Not at All in Agreement			Totally Agree			
Socio-Demographic Variable	Total	1, *N* = 11 ^1^	2, *N* = 29 ^1^	3, *N* = 69 ^1^	4, *N* = 112 ^1^	5, *N* = 53 ^1^	6, *N* = 16 ^1^	*p*-Value ^2^
** *Type* **								
Total	286	10 (3.5%)	29 (10%)	67 (23%)	112 (39%)	53 (19%)	15 (5.2%)	0.075
Female	165	5 (3.0%)	22 (13%)	43 (26%)	63 (38%)	23 (14%)	9 (5.5%)	
Male	121	5 (4.1%)	7 (5.8%)	24 (20%)	49 (40%)	30 (25%)	6 (5.0%)	
** *Age Category* **								
Total	290	11 (3.8%)	29 (10%)	69 (24%)	112 (39%)	53 (18%)	16 (5.5%)	0.079
18–24	142	5 (3.5%)	20 (14%)	35 (25%)	50 (35%)	25 (18%)	7 (4.9%)	
25–30	106	3 (2.8%)	7 (6.6%)	24 (23%)	40 (38%)	26 (25%)	6 (5.7%)	
31–36	42	3 (7.1%)	2 (4.8%)	10 (24%)	22 (52%)	2 (4.8%)	3 (7.1%)	
** *Language Region* **								
Total	290	11 (3.8%)	29 (10%)	69 (24%)	112 (39%)	53 (18%)	16 (5.5%)	0.6
German-speaking Switzerland	79	2 (2.5%)	8 (10%)	17 (22%)	37 (47%)	11 (14%)	4 (5.1%)	
Latin Switzerland	211	9 (4.3%)	21 (10%)	52 (25%)	75 (36%)	42 (20%)	12 (5.7%)	
** *Level of Training* **								
Total	290	11 (3.8%)	29 (10%)	69 (24%)	112 (39%)	53 (18%)	16 (5.5%)	0.1
High School	49	2 (4.1%)	9 (18%)	10 (20%)	22 (45%)	4 (8.2%)	2 (4.1%)	
Bachelor	97	3 (3.1%)	8 (8.2%)	31 (32%)	33 (34%)	17 (18%)	5 (5.2%)	
Master	108	3 (2.8%)	9 (8.3%)	19 (18%)	43 (40%)	25 (23%)	9 (8.3%)	
PhD	11	1 (9.1%)	0 (0%)	3 (27%)	7 (64%)	0 (0%)	0 (0%)	
Other	25	2 (8.0%)	3 (12%)	6 (24%)	7 (28%)	7 (28%)	0 (0%)	
** *Field of Study* **								
Total	290	11 (3.8%)	29 (10%)	69 (24%)	112 (39%)	53 (18%)	16 (5.5%)	0.073
Law	4	0 (0%)	1 (25%)	2 (50%)	0 (0%)	0 (0%)	1 (25%)	
Economy, Trade, Management and Services	26	0 (0%)	3 (12%)	9 (35%)	7 (27%)	2 (7.7%)	5 (19%)	
Engineering, Architecture	40	2 (5.0%)	2 (5.0%)	10 (25%)	14 (35%)	10 (25%)	2 (5.0%)	
Pedagogy or Education	6	0 (0%)	0 (0%)	3 (50%)	3 (50%)	0 (0%)	0 (0%)	
Nat. Sci., Life Sci., and Computer Sci.	71	4 (5.6%)	8 (11%)	19 (27%)	29 (41%)	10 (14%)	1 (1.4%)	
Health Sciences	94	1 (1.1%)	11 (12%)	19 (20%)	38 (40%)	19 (20%)	6 (6.4%)	
Humanities and Social Sciences	24	2 (8.3%)	1 (4.2%)	4 (17%)	13 (54%)	3 (12%)	1 (4.2%)	
Other	25	2 (8.0%)	3 (12%)	3 (12%)	8 (32%)	9 (36%)	0 (0%)	
** *Nationality* **								
Total	290	11 (3.8%)	29 (10%)	69 (24%)	112 (39%)	53 (18%)	16 (5.5%)	0.13
Germany	5	0 (0%)	1 (20%)	0 (0%)	4 (80%)	0 (0%)	0 (0%)	
England	1	0 (0%)	0 (0%)	1 (100%)	0 (0%)	0 (0%)	0 (0%)	
Brazil	8	1 (12%)	2 (25%)	2 (25%)	3 (38%)	0 (0%)	0 (0%)	
Cameroon	1	0 (0%)	0 (0%)	0 (0%)	1 (100%)	0 (0%)	0 (0%)	
Colombia	1	0 (0%)	0 (0%)	0 (0%)	1 (100%)	0 (0%)	0 (0%)	
Spain	5	0 (0%)	2 (40%)	0 (0%)	2 (40%)	0 (0%)	1 (20%)	
France	23	0 (0%)	0 (0%)	5 (22%)	5 (22%)	11 (48%)	2 (8.7%)	
Greece	1	0 (0%)	0 (0%)	1 (100%)	0 (0%)	0 (0%)	0 (0%)	
Iran	1	0 (0%)	0 (0%)	1 (100%)	0 (0%)	0 (0%)	0 (0%)	
Italy	2	0 (0%)	0 (0%)	1 (50%)	1 (50%)	0 (0%)	0 (0%)	
Liechtenstein	1	0 (0%)	0 (0%)	1 (100%)	0 (0%)	0 (0%)	0 (0%)	
Philippines	1	0 (0%)	0 (0%)	0 (0%)	1 (100%)	0 (0%)	0 (0%)	
Portugal	7	0 (0%)	0 (0%)	2 (29%)	5 (71%)	0 (0%)	0 (0%)	
DRC ^3^	1	0 (0%)	0 (0%)	0 (0%)	1 (100%)	0 (0%)	0 (0%)	
Switzerland	231	9 (3.9%)	24 (10%)	55 (24%)	88 (38%)	42 (18%)	13 (5.6%)	
Vietnam	1	1 (100%)	0 (0%)	0 (0%)	0 (0%)	0 (0%)	0 (0%)	

^1^ n (%); ^2^ Chi-square/Fisher test (depending on n); ^3^ Democratic Republic of the Congo.

**Table 6 foods-11-02771-t006:** Acceptability of consuming insect-based foods by behavioral variables.

		Not at All in Agreement			Totally Agree			
Behavioral Variables	Total	1	2	3	4	5	6	*p*-Value ^2^
		*N* = 11 ^1^	*N* = 29 ^1^	*N* = 69 ^1^	*N* = 112 ^1^	*N* = 53 ^1^	*N* = 16 ^1^	
**Visible Insects**								**<0.001**
1	104 (36%)	11 (100%)	28 (97%)	33 (48%)	25 (22%)	5 (9.4%)	2 (12%)	
2	70 (24%)	0 (0%)	1 (3.4%)	25 (36%)	33 (29%)	9 (17%)	2 (12%)	
3	60 (21%)	0 (0%)	0 (0%)	10 (14%)	35 (31%)	14 (26%)	1 (6.2%)	
4	35 (12%)	0 (0%)	0 (0%)	1 (1.4%)	17 (15%)	16 (30%)	1 (6.2%)	
5	14 (4,8%)	0 (0%)	0 (0%)	0 (0%)	1 (0.9%)	8 (15%)	5 (31%)	
6	7 (2.4%)	0 (0%)	0 (0%)	0 (0%)	1 (0.9%)	1 (1.9%)	5 (31%)	
**Insects Not Visible**								**<0.001**
1	52 (18%)	10 (91%)	22 (76%)	14 (20%)	6 (5.4%)	0 (0%)	0 (0%)	
2	59 (20%)	0 (0%)	7 (24%)	21 (30%)	27 (24%)	4 (7.5%)	0 (0%)	
3	90 (31%)	1 (9.1%)	0 (0%)	26 (38%)	45 (40%)	15 (28%)	3 (19%)	
4	62 (21%)	0 (0%)	0 (0%)	7 (10%)	28 (25%)	23 (43%)	4 (25%)	
5	20 (6.9%)	0 (0%)	0 (0%)	1 (1.4%)	5 (4.5%)	9 (17%)	5 (31%)	
6	7 (2.4%)	0 (0%)	0 (0%)	0 (0%)	1 (0.9%)	2 (3.8%)	4 (25%)	
**Meat = Health**								**<0.001**
1	25 (8.6%)	4 (36%)	6 (21%)	4 (5.8%)	5 (4.5%)	4 (7.5%)	2 (12%)	
2	61 (21%)	1 (9.1%)	9 (31%)	11 (16%)	22 (20%)	14 (26%)	4 (25%)	
3	58 (20%)	1 (9.1%)	2 (6.9%)	10 (14%)	29 (26%)	14 (26%)	2 (12%)	
4	94 (32%)	0 (0%)	5 (17%)	27 (39%)	40 (36%)	16 (30%)	6 (38%)	
5	35 (12%)	1 (9.1%)	3 (10%)	14 (20%)	11 (9.8%)	5 (9.4%)	1 (6.2%)	
6	17 (5.9%)	4 (36%)	4 (14%)	3 (4.3%)	5 (4.5%)	0 (0%)	1 (6.2%)	
**Meat Lovers**								0.057
1	25 (8.6%)	5 (45%)	4 (14%)	7 (10%)	7 (6.2%)	1 (1.9%)	1 (6.2%)	
2	30 (10%)	1 (9.1%)	5 (17%)	10 (14%)	10 (8.9%)	2 (3.8%)	2 (12%)	
3	56 (19%)	0 (0%)	7 (24%)	9 (13%)	27 (24%)	12 (23%)	1 (6.2%)	
4	95 (33%)	4 (36%)	8 (28%)	22 (32%)	35 (31%)	21 (40%)	5 (31%)	
5	60 (21%)	0 (0%)	3 (10%)	16 (23%)	24 (21%)	12 (23%)	5 (31%)	
6	24 (8.3%)	1 (9.1%)	2 (6.9%)	5 (7.2%)	9 (8.0%)	5 (9.4%)	2 (12%)	
**Food Neophobia**								**0.004**
2	4 (1.4%)	0 (0%)	0 (0%)	3 (4.3%)	1 (0.9%)	0 (0%)	0 (0%)	
3	23 (7.9%)	2 (18%)	5 (17%)	7 (10%)	6 (5.4%)	2 (3.8%)	1 (6.2%)	
4	74 (26%)	5 (45%)	5 (17%)	20 (29%)	33 (29%)	8 (15%)	3 (19%)	
5	106 (37%)	2 (18%)	13 (45%)	30 (43%)	39 (35%)	16 (30%)	6 (38%)	
6	83 (29%)	2 (18%)	6 (21%)	9 (13%)	33 (29%)	27 (51%)	6 (38%)	
**Neophobia of New Technology**								**<0.001**
1	6 (2.1%)	2 (18%)	0 (0%)	0 (0%)	1 (0.9%)	0 (0%)	3 (19%)	
2	31 (11%)	4 (36%)	1 (3.4%)	10 (14%)	11 (9.8%)	2 (3.8%)	3 (19%)	
3	56 (19%)	1 (9.1%)	12 (41%)	15 (22%)	14 (12%)	14 (26%)	0 (0%)	
4	118 (41%)	2 (18%)	9 (31%)	27 (39%)	52 (46%)	21 (40%)	7 (44%)	
5	57 (20%)	2 (18%)	6 (21%)	15 (22%)	23 (21%)	10 (19%)	1 (6.2%)	
6	22 (7.6%)	0 (0%)	1 (3.4%)	2 (2.9%)	11 (9.8%)	6 (11%)	2 (12%)	
**Interest in Health**								0.5
1	3 (1.0%)	0 (0%)	2 (6.9%)	1 (1.4%)	0 (0%)	0 (0%)	0 (0%)	
2	21 (7.2%)	0 (0%)	2 (6.9%)	10 (14%)	6 (5.4%)	3 (5.7%)	0 (0%)	
3	15 (5.2%)	0 (0%)	1 (3.4%)	2 (2.9%)	7 (6,2%)	4 (7.5%)	1 (6.2%)	
4	126 (43%)	8 (73%)	14 (48%)	24 (35%)	48 (43%)	22 (42%)	10 (62%)	
5	51 (18%)	1 (9.1%)	3 (10%)	16 (23%)	20 (18%)	9 (17%)	2 (12%)	
6	74 (26%)	2 (18%)	7 (24%)	16 (23%)	31 (28%)	15 (28%)	3 (19%)	
**Practicality**								0.5
2	1 (0.3%)	0 (0%)	0 (0%)	1 (1.4%)	0 (0%)	0 (0%)	0 (0%)	
3	39 (13%)	2 (18%)	5 (17%)	7 (10%)	12 (11%)	9 (17%)	4 (25%)	
4	194 (67%)	6 (55%)	20 (69%)	46 (67%)	77 (69%)	35 (66%)	10 (62%)	
5	54 (19%)	2 (18%)	4 (14%)	15 (22%)	23 (21%)	8 (15%)	2 (12%)	
6	2 (0.7%)	1 (9.1%)	0 (0%)	0 (0%)	0 (0%)	1 (1.9%)	0 (0%)	
**Value for Money**								0.2
1	3 (1.0%)	1 (9.1%)	1 (3.4%)	1 (1.4%)	0 (0%)	0 (0%)	0 (0%)	
2	40 (14%)	0 (0%)	4 (14%)	12 (17%)	17 (15%)	7 (13%)	0 (0%)	
3	80 (28%)	4 (36%)	8 (28%)	16 (23%)	34 (30%)	13 (25%)	5 (31%)	
4	103 (36%)	4 (36%)	11 (38%)	27 (39%)	40 (36%)	17 (32%)	4 (25%)	
5	54 (19%)	2 (18%)	4 (14%)	12 (17%)	20 (18%)	12 (23%)	4 (25%)	
6	10 (3.4%)	0 (0%)	1 (3.4%)	1 (1.4%)	1 (0.9%)	4 (7.5%)	3 (19%)	

^1^ n (%); ^2^ Fisher test. Bold in this case indicated that these *p*-values were significant.

**Table 7 foods-11-02771-t007:** Comparison of variables composing the acceptance score (ACI) obtained in the present study and those obtained by Schlup and Brunner, based on the participants’ responses [23].

Argument		Mean Score (This Study)	SD	Mean Score (Schlup and Brunner)	SD (Schlup and Brunner)	T-Statistics	Degrees of Freedom	*p*-Value
Sustainability	Extent of agreement	4.8	1.4	4	1.7	6.67	663.38	**<0.001**
	Perceived relevance	3.7	1.7	2.8	1.6	6.96	602.07	**<0.001**
Health	Extent of agreement	4.4	1.3	3.9	1.6	4.46	664.58	**<0.001**
	Perceived relevance	3.4	1.7	2.9	1.6	3.87	602.07	**<0.001**
Price	Extent of agreement	4.4	1.4	3.7	1.6	6.02	655.07	**<0.001**
	Perceived relevance	3.3	1.7	2.5	1.5	6.34	578.84	**<0.001**
Taste	Extent of agreement	4	1.6	3.6	1.6	3.2	622.03	**<0.001**
	Perceived relevance	3.4	1.7	2.8	1.6	4.64	602.07	**<0.001**
Trend	Extent of agreement	3.2	1.5	3.3	1.5	−0.85	622.03	0.39
	Perceived relevance	2.8	1.6	2.6	1.5	1.65	600.7	0.1

Bold in this case indicated that these *p*-values were significant.

## Data Availability

The data presented in this study are available on request from the corresponding author. The data are not publicly available due to due privacy and data protections issues.

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
