# Peer review of "The Consumption of Insects in Switzerland: University-Based Perspectives of Entomophagy"

_foods, 2022, doi:10.3390/foods11182771_

Round 1
Reviewer 1 Report
This article aimed at evaluating the level of acceptability of entomophagy (insects as a food) among young adults in a Swiss university. The article is well written with appropriate introduction, methodology, results and discussion. However, the manuscript could be further proofread to eliminate inconsistencies in the tables and to reduce grammatical errors.
Some minor comments:
Line 46: Soil consumption. Please clarify. Does the author mean the rearing substrate or area required for insect rearing?
Lines 65: Replace “(mealworm)” with “(yellow mealworm)” for clarification.
Tenebrio molitor is the yellow mealworm.
Line 66: Replace “(cricket)” with “(house cricket)”
Acheta domesticus is the house cricket.
Lines 542-543: Supplementary Materials link unavailable.
Please attach questionnaire “Table S1: Survey: Consumption of insect-based foods in Switzerland” to the link in supplementary materials as initially referenced in line 121 of the manuscript.
Author Response
We appreciate the reviewer’s comments and believe to have answered the comments sufficiently below for the paper to be reconsidered for publication in Foods:
Reviewer 1:
This article aimed at evaluating the level of acceptability of entomophagy (insects as a food) among young adults in a Swiss university. The article is well written with appropriate introduction, methodology, results and discussion. However, the manuscript could be further proofread to eliminate inconsistencies in the tables and to reduce grammatical errors.
The manuscript has been carefully proofread to eliminate inconsistencies in the tables and to reduce grammatical errors.
Some minor comments:
Line 46: Soil consumption. Please clarify. Does the author mean the rearing substrate or area required for insect rearing?
Clarified (“soil utilization for rearing”)
Lines 65: Replace “(mealworm)” with “(yellow mealworm)” for clarification. Tenebrio molitor is the yellow mealworm.
Replaced with yellow mealworm
Line 66: Replace “(cricket)” with “(house cricket)” Acheta domesticus is the house cricket.
Replaced with “house cricket”
Lines 542-543: Supplementary Materials link unavailable.
Please attach questionnaire “Table S1: Survey: Consumption of insect-based foods in Switzerland” to the link in supplementary materials as initially referenced in line 121 of the manuscript.
Table S1: Survey: Consumption of insect-based foods in Switzerland has been added
Reviewer 2 Report
The paper aims to assess acceptability in consuming insects (ACI) in relation to each respondent's perception of entomophagy. The paper is adequately developed both in terms of the organization of the paragraphs and how they are developed. The only comment I would like to make relates to the paragraph on the questionnaire employed. Normally in this type of work, where the data collected with the questionnaire is used describe and measure three constructs (in this case Q1; Q2; Q3 page 3) one needs to indicate how the items and their scales were derived. I remind you that the measurement scales must be validated by the scientific literature.
Author Response
We appreciate the reviewer’s comments and believe to have answered the comments sufficiently below for the paper to be reconsidered for publication in Foods:
The paper aims to assess acceptability in consuming insects (ACI) in relation to each respondent's perception of entomophagy. The paper is adequately developed both in terms of the organization of the paragraphs and how they are developed. The only comment I would like to make relates to the paragraph on the questionnaire employed. Normally in this type of work, where the data collected with the questionnaire is used describe and measure three constructs (in this case Q1; Q2; Q3 page 3) one needs to indicate how the items and their scales were derived. I remind you that the measurement scales must be validated by the scientific literature.
The questionnaire and it’s has been validated previously by “Schlup, Y.; Brunner, T. Prospects for insects as food in Switzerland: A tobit regression. Food Qual Prefer 2018, 64, 37‑46.” This was elaborated in line 172-173 “The questionnaire used in this study is an adaptation of the one developed and vali-dated by Schlup and Brunner [23]”